# Spin-torque skyrmion resonance in a frustrated magnet

Nirel Bernstein ®[1], Hang Li[2,3], Benjamin Assouline ®[1], Yong-Chang Lau[3,4], Igor Rozhansky ®[1,5], Wenhong Wang[2] & Amir Capua ®[1]✉

The frustrated $Fe_3Sn_2$ magnet is technologically attractive due to its extreme-temperature skyrmion stability, large topological Hall effect, and current-induced helicity switching attributed to a self-induced spin-torque. Here, we present a current-driven skyrmion resonance technique excited by self-induced spin-torque in $Fe_3Sn_2$. The dynamics are probed optically in a time-resolved measurement enabling us to distinguish between the excited modes. We find that only the breathing and rotational counterclockwise modes are excited, rather than the three modes typically observed in Dzyaloshinskii-Moriya interaction-dominated magnetic textures. When a DC current is passed through the crystal, the skyrmion resonance linewidth is modulated. Our micromagnetic simulations indicate that the linewidth broadening arises from an effective damping-like spin-orbit torque. Accordingly, we extract an effective spin Hall conductivity of $\sim 793 \pm 176 \, (\hbar/e) \, (\Omega \, cm)^{-1}$. Complementary planar Hall measurements suggest a small yet finite contribution of the real-space spin texture in the electronic transport in addition to a primary $k$-space contribution. Our results bring new insights into the anisotropic nature of spin-torques in frustrated magnets and to the possibility of using the skyrmion resonance as a sensor for spin currents.

Magnetic skyrmions are attractive candidates for dense data distribution and storage applications. Conventionally, Dzyaloshinskii-Moriya interaction (DMI) is responsible for the skyrmion's topological protection. Recently, it was predicted that also magnetic frustration in ferromagnetic (FM) crystals can provide the topological protection of the skyrmion instead of DMI[1–4] and the effect was soon discovered in $Fe_3Sn_2$ bulk crystals[5–8]. $Fe_3Sn_2$ is a technologically promising material system that serves as an excellent platform for exploring the role of spin chirality. The interplay between the exchange interaction, uniaxial magnetic anisotropy (UMA), and dipole-dipole interactions gives rise to a variety of spin configurations including stripes, bubbles, and skyrmions that have displayed extreme thermal stability of up to $\sim 600 \, K^{3–5}$. Interestingly, the electronic charge transport in $Fe_3Sn_2$ is strongly affected by the topologies of the band structure and of the real-space spin texture. This is seen by the pronounced topological[9,10]

and anomalous Hall effects (THE and AHE, respectively)[11,12]. Furthermore, recent experiments showed that electrical currents can drive the motion of skyrmions in $Fe_3Sn_2$ and even switch their helicity and topological charge $Q$[7,13,14]. The switching effect was attributed to a self-induced spin-torque that is assisted by Joule heating[3,14], yet the underlying mechanism is still debated.

Current-driven dynamics of skyrmions in bulk crystals are usually excited through spin-transfer torques (STT) by spin-polarized electric currents that interact with the localized spins[15]. However, $Fe_3Sn_2$ exhibits a unique combination of a layered kagome crystal structure with magnetic frustration and a non-vanishing spin-orbit coupling (SOC) which may give rise to an additional spin-orbit torque (SOT). SOC is well known to affect the $k$-space band topology leading to the spin Hall effect (SHE)[16,17]. Additionally, in layered materials, vertical spin currents can arise from interfacial spin-orbit interactions. Often,

[1]Institute of Applied Physics, The Hebrew University of Jerusalem, Jerusalem, Israel. [2]School of Electronics and Information Engineering, Tiangong University, Tianjin, China. [3]Beijing National Laboratory for Condensed Matter Physics, Institute of Physics, Chinese Academy of Sciences, Beijing, China. [4]University of Chinese Academy of Sciences, Beijing, China. [5]National Graphene Institute, University of Manchester, Manchester, UK. ✉e-mail: amir.capua@mail.huji.ac.il

the role of SOC in determining the band topology can be substituted by the spin chirality of the magnetic texture[18]. This is the origin of the momentum ($k$) space contribution to the THE where the Berry phase in momentum space is produced by the spin chirality of the localized spin structures such as skyrmions[19–21]. Alternatively, the real-space magnetic topology of the localized spin structures can also cause the THE. The underlying microscopic mechanism of the real-space spin texture contribution to the THE was described by spin separation which arises from the opposite polarity of the emergent magnetic field for opposite spins[21–24]. Therefore, in addition to the transverse charge current that arises when spin populations are uneven, e.g., as in FMs, this was also predicted to result in a spin current[22,24–26]. Figure 1a schematically illustrates these spin and charge transport phenomena. The multitude of mechanisms capable of generating spin-torque highlights the technological relevance of $Fe_3Sn_2$ while also posing a major challenge in unlocking its full potential.

Microwave resonance experiments are pivotal for probing spin-torques. However, in textured magnets, the Slonczewski spin-torque becomes highly anisotropic rendering its effect unclear. Furthermore, the complexity is aggravated by the relatively less explored nature of skyrmion resonance dynamics. Early theoretical studies of the skyrmion resonance dynamics predicted the excitation of two fundamental in-plane (IP) rotational clockwise (CW) and counter-clockwise (CCW) modes and a third out-of-plane (OOP) breathing mode[27]. These modes were measured immediately afterwards in a cubic helimagnet[28]. In the cubic metallic, semiconducting, and insulating chiral magnets such as MnSi, $Fe_{1-x}Co_xSi$, and $Cu_2OSeO_3$, respectively, it was shown that the dynamics of magnetic textures follow universal laws determined solely by the chiral and critical field energies[29]. In contrast, the strong UMA in the rhombohedral $GaV_4S_8$ chiral magnet was shown to interchange the hierarchy of the excitations[30]. The UMA was also responsible for annihilating the CW mode in multilayered thin films where the skyrmions are stabilized by interfacial DMI and UMA[31,32]. Interestingly, in frustrated triangular magnets, it has been predicted that UMA can strongly influence the spin ordering, leading to a complex resonance spectrum that is accompanied by oscillations of the electric polarization[1]. The complexity of the dynamics was attributed to the nonlinear coupling between the additional helicity degree of freedom present in frustration-induced magnetic skyrmions and the skyrmion's center of mass. Furthermore, the study also predicted that only the CW and breathing modes can be excited in such a system. Despite the significant physical insights that can be captured by studying the skyrmion resonance in frustrated magnets, the empirical validation of the predicted dynamics remains very limited.

Primarily, the skyrmion resonance dynamics were explored using broadband microwave absorption spectroscopy[28–32]. However, it is well known that the ferromagnetic resonance (FMR) can also be excited by AC spin-torques. This is known as the spin-torque FMR (STFMR) which proved highly effective for exploring SOTs[33]. To this end, a FM is deposited on the material of interest and serves as a spin current sensor by measuring the current-induced FMR linewidth modulation. However, in textured magnets, the additional FM is bound to suppress the texture, potentially eliminating it, and as a result deform the topologies of both the energy bands and of the spin texture in $k$- and real-spaces, respectively.

Here, we report a time-resolved optically probed spin-torque driven skyrmion resonance technique in which the dynamics are excited by a self-induced RF spin-torque in a fashion that is reminiscent of the STFMR technique[33]. We apply the technique to the frustrated $Fe_3Sn_2$ crystal and find that indeed only the breathing and CCW skyrmion modes are excited in accord with theoretical predictions. The dynamical responses illustrate the magnetic phase transitions from a disordered state to a stripe phase and finally to an ordered lattice phase as the externally applied magnetic field is increased. Interestingly, when a DC charge current is passed through the crystal, we observe that the linewidth of each mode is modulated depending on the magnetic phase. Micromagnetic time-domain numerical simulations that account for the STT and SOT indicate that while the AC STT and AC SOT are both capable of exciting the skyrmion resonance dynamics, only the DC SOT is capable of modulating the linewidth. Complementary transport measurements reveal a primary $k$-space and a small yet finite real-space contribution to the Hall and planar Hall effects (PHE) highlighting a possible connection between the spin texture and the damping-like torque. The experimental technique is thus key for the understanding of spin dynamics in textured magnets and for developing relevant applications in frustrated magnets.

## Results

The $Fe_3Sn_2$ crystal has a rhombohedral unit cell which belongs to the $R\bar{3}m$ space group. It consists of alternating double kagome Fe-Sn and hexagonal Sn layers stacked along the c-axis (Fig. 1b) having lattice constants of $a = b = 5.338$ Å and $c = 19.789$ Å in hexagonal coordinate system[34]. The adjacent kagome Fe-Sn layers are offset, and each consists of two different Fe-Fe bond lengths of 2.732 Å and 2.582 Å. Crystals of $Fe_3Sn_2$ were synthesized by chemical vapor transport reaction as described in 'Methods'. Figure 1c presents a scanning electron microscope (SEM) image of the crystal.

The magnetization dynamics were studied using an optically probed STFMR technique (OSTFMR)[33,35,36]. An RF charge current of frequency $f$ was passed through the crystal and excited the dynamics. The responses were probed using a femtosecond laser that was phase-locked to the RF signal. The OOP component of the RF magnetization, $m_z$, was probed using the magneto-optical Kerr effect (MOKE) and an optical delay line provided the temporal resolution. The current was passed along the $\hat{x}$ direction corresponding to the [100] crystal axis and the external magnet field $\vec{H}$ of magnitude $H_0$ was applied in the sample plane at $\theta_H = 30°$ with respect to the [100] axis as indicated in Fig. 1c (see 'Methods' for further details).

An example of an OSTFMR trace at 14 $GHz$ is presented in Fig. 1d. The trace presents $m_z(t)$ as a function of $H_0$ in which a resonance peak appears at $\sim 300\,mT$. The extracted phase response, $\phi(H_0)$, is represented by the black dashed line. Interestingly, $\phi(H_0)$ reveals a total phase shift, $\Delta\phi$, of $\sim 2\pi$ accumulated across the resonance rather than the expected $\pi$ shift, indicating that actually two resonance modes were measured. This is confirmed by plotting the amplitude response $|m_z(H_0)|$ presented in Fig. 1e that reveals a secondary feature at $\sim 250\,mT$. Namely, the AC magnetic susceptibility consists of two Lorentzian lines $\chi_m = \sum_{i=1,2} A_i \cdot \frac{1}{\left(H_{res_i}^2 - H_0^2\right) + iH_0\Delta H_i} \cdot e^{i\phi_i}$ where $A_i$, $H_{res_i}$, and $\Delta H_i$ are the amplitude, resonance field, and linewidth of each resonance mode, and $\phi_i$ is the phase relative to the RF excitation. The reconstructed $m_z(H_0)$ is presented as well in Fig. 1e together with the two retrieved modes and agrees well with the measurement. Using micromagnetic simulations that account for the exchange, dipole-dipole, tilted UMA, and Zeeman energies (see 'Methods'), we find that the lower (higher) resonance field corresponds to the CCW (breathing) mode. In the general case, the CW mode should also be excited, however, it is typically of a smaller amplitude while the combined effect of frustration and strong exchange interaction further suppress its excitation[1]. The driving RF torque can stem from the RF Oersted or RF spin-torque. In frustrated magnets, an IP RF Oersted field can only drive the CCW mode[1,27]. In contrast, the RF spin-torque consists of IP and OOP components that can excite also the translational mode. Namely, the excitation of the two orthogonal modes implies the generation of RF spin-torque in the crystal. The negligible self-induced Oersted torque[33] was verified by passing a DC current density $\vec{J}_c$ of magnitude $J_c$ which had no effect on $H_{res_i}$ as seen in Fig. 1f.

The same analysis was repeated for additional frequencies. The top panels of Fig. 2a show characteristic spectra measured at 6, 8, 12,

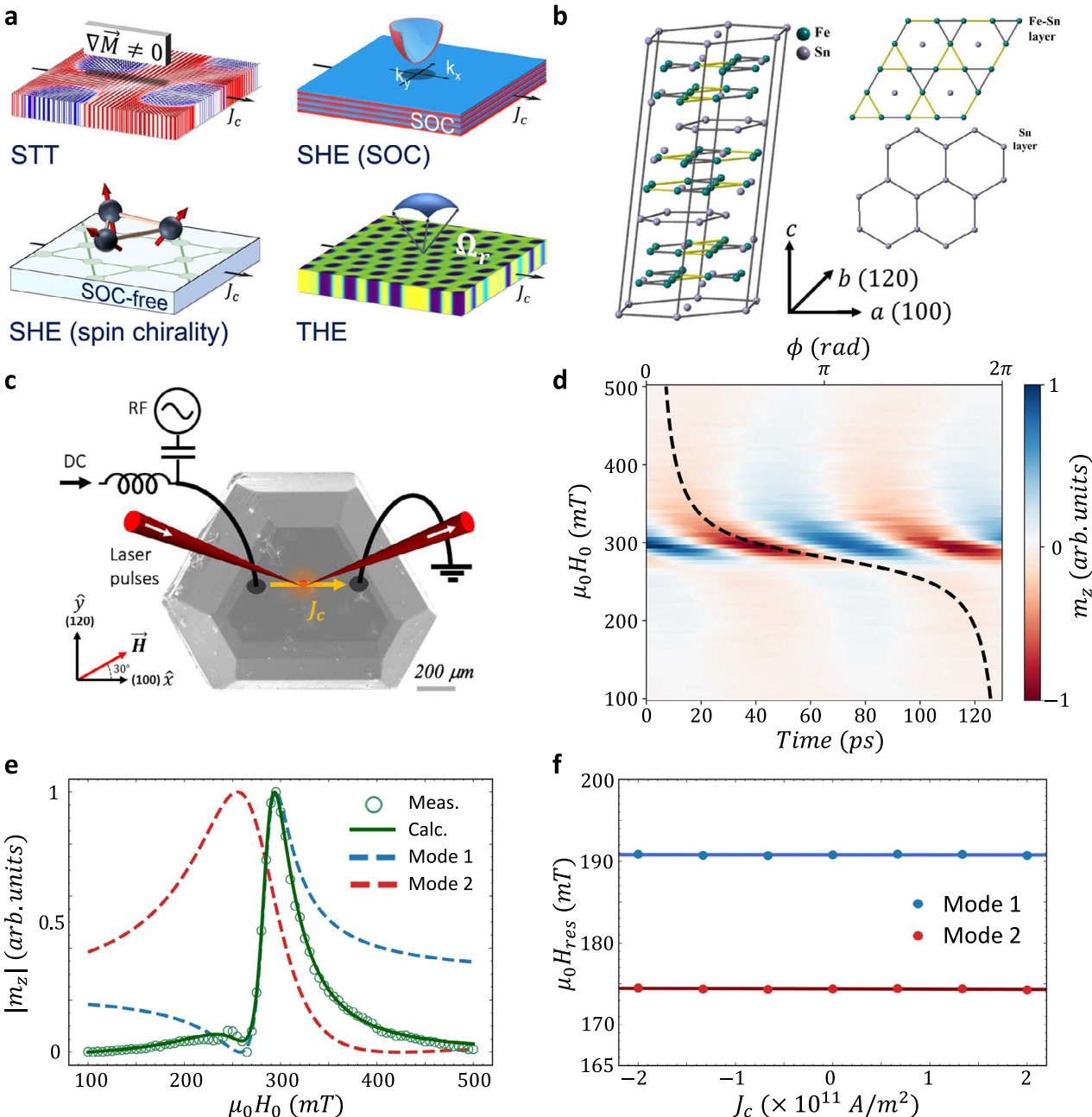

**Fig. 1 | Characteristic OSTFMR measurement. a** Schematic illustration of charge and spin transport mechanisms in Fe$_3$Sn$_2$: spin-polarized electric currents responsible for STT, SOC-based bulk and interfacial SHE, THE induced by $k$-space band topology emerging from spin-chirality, and THE induced by real-space topology of localized spin structures. **b** Crystal structure of Fe$_3$Sn$_2$. **c** SEM image of the Fe$_3$Sn$_2$ crystal together with a schematic of the experimental technique. RF currents are driven through the crystal and excite the skyrmion resonance. $m_z$ is detected in a polar-MOKE configuration using a phase-locked femtosecond laser. **d** OSTFMR trace of $m_z(t, H_0)$ at $14\,GHz$. Black dashed line indicates $\phi$ vs. $H_0$. **e** Measured amplitude response $|m_z(H_0)|$ (green open circles) together with the two fitted modes (red and blue dashed lines) and the reconstructed response (green solid line). **f** Extracted $H_{res}$ at $10\,GHz$ for different values of $J_c$. $J_c$ was estimated using the Poisson equation while accounting for the geometrical dimensions of the crystal (details in Supplementary Note 2). In (**d**, **e**) $m_z$ was normalized to its peak value.

and $14\,GHz$. At high frequencies, distinct resonances appear whereas at low frequencies the spectrum is random. This behavior is summarized by plotting the $f - H_{res_i}$ dispersion relations presented in the main panel of the figure. A phase transition is seen at $\sim 10\,GHz$ ($\sim 175\,mT$). This transition can be understood from the dynamical micromagnetic simulations. To this end, the static textures were first determined and are presented in the top panels of Fig. 2b. A disordered phase is seen at $H_0 = 0$ that evolves into a stripe phase at $75\,mT$ after which an ordered lattice phase emerges. When $H_0$ is further increased, the texture

gradually saturates. The dynamical responses were obtained from the Fourier transform of the impulse response of the mean OOP magnetization $M_z$ and are presented in the main panel of Fig. 2b. The calculated dynamics correlate with the magnetic phase transitions and reproduce the trends seen in the measurements: scattered resonance peaks measured at $\mu_0 H_0 < 150\,mT$ and a homogeneous monotonic dispersion curve consisting of two modes at high $H_0$. Indications of the stripe phase are also found in the measurement as seen in the spectrum of $8\,GHz$ which is significantly broader due to the higher

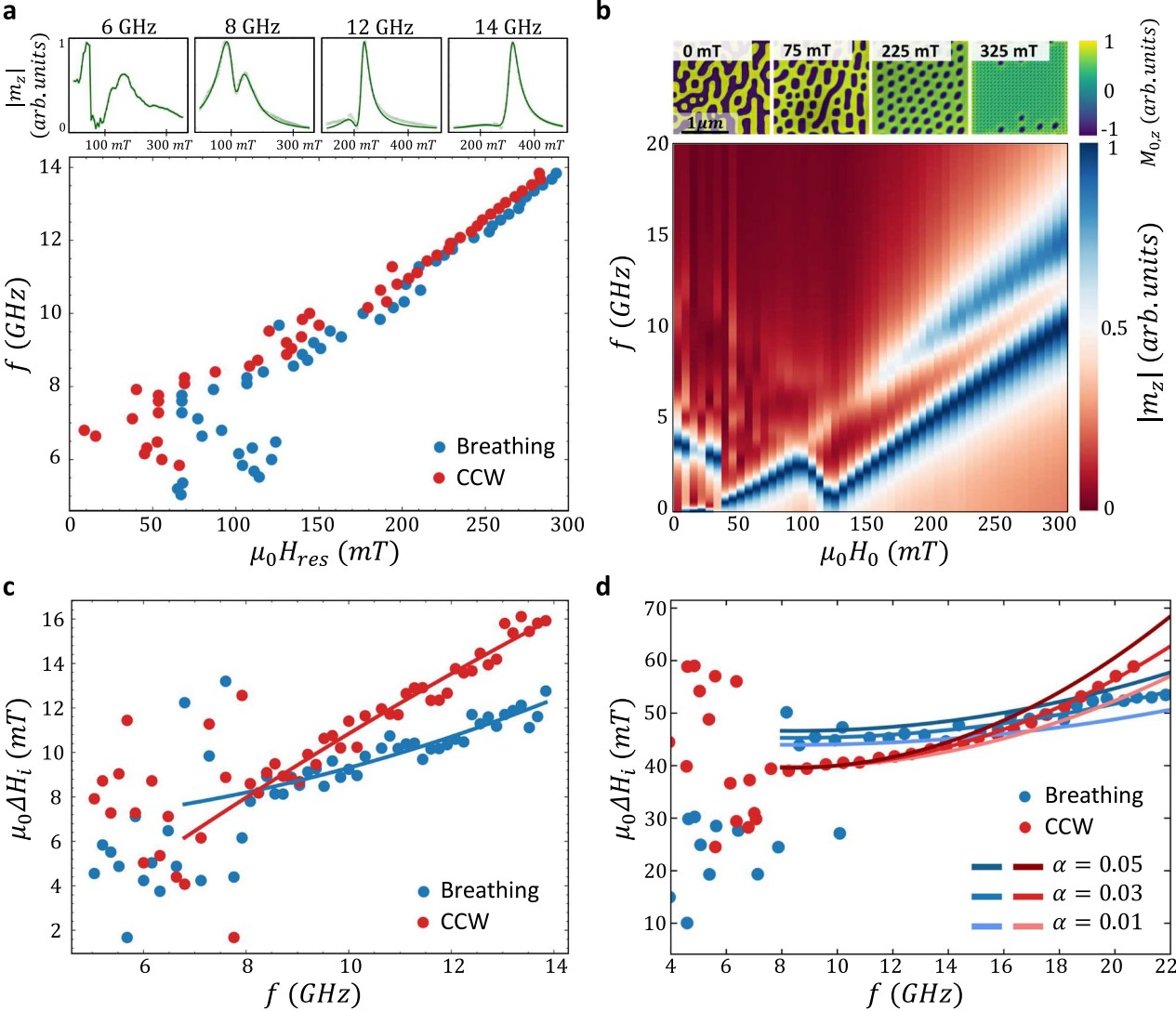

**Fig. 2 | Measured and calculated resonance dynamics of Fe₃Sn₂. a** Measured dynamical response. Top panels present measured $|m_z|$ as a function of $H_0$ at 6, 8, 12, and 14 $GHz$ (light green open circles). Reconstructed responses are indicated in green solid line. Data is presented in normalized units. Main panel: Frequency dispersion relations of the breathing and CCW resonance modes. **b** Micromagnetic dynamical simulations. Top panels present the calculated textures at $\mu_0 H_0 = 0$, 75, 225, and 325 $mT$. The textures are represented by plotting the equilibrium OOP component of the magnetization, $M_{0,z}$. The main panel presents the calculated frequency response of $|m_z|$ as a function of $H_0$ for $\alpha = 0.05$. Each response was normalized to its peak value. **c** Measured resonance linewidths ($\Delta H_i$) of CCW and breathing modes as a function of $f$. Solid lines represent a fit to a second order polynomial function. **d** Calculated $\Delta H_i$ of the two modes. Solid lines illustrate the calculation for $\alpha = 0.01$, 0.03, and 0.05. In (**a**, **c**, **d**) data of the breathing and CCW modes are indicated by blue and red colors, respectively.

motional degrees of freedom. From the calculations, $Q = 0$ was determined in the lattice phase (see Supplementary Note 3). Following the conventional terminology, we use the term skyrmion resonance although the magnetic texture is topologically trivial.

Interestingly, features appearing in the measured resonance linewidths are also reproduced in the calculation. Figure 2c presents the measured $\Delta H_i$ as a function of $f$. It is seen that $\Delta H$ of the CCW mode increases more rapidly with $f$ as compared to the breathing mode. The same behavior is seen in the calculated $\Delta H_i$ presented in Fig. 2d which was obtained using $\Delta H_i = \left(\frac{\partial f_{res}}{\partial H_0}\right)^{-1} \Delta f_i$. The overall broader calculated $\Delta H_i$ stems from the inhomogeneous broadening induced by the interfaces of the finite calculation space. A Gilbert damping of $\alpha = 0.032 \pm 0.008$ was evaluated from the $f$ dependent $\Delta H$ broadening of the CCW mode in the range $f > 10\,GHz$ which is comparable to previously measured values in thin films of Fe₃Sn₂[37]. In the CCW mode, the spins undergo a local precessional trajectory about an effective field. Therefore, the Rayleigh viscous-like damping process

manifests in the measured quasi-linear dependence at high $f$. In contrast, in the breathing mode, the spins follow a non-conical expansion and contraction-type motion that results in a weaker dependence of $\Delta H$ on $f$.

Next, we examine the influence of $J_c$ on the skyrmion resonance linewidth. In the following experiments, the current was passed along the [100] axis. Figure 3a–c present the measured data for the disordered, stripe, and lattice phases by plotting the variation of $\Delta H_i$ relative to the zero-bias linewidth $\Delta H_i^0$. In the disordered phase, a significant random modulation of $\Delta H_{1,2}$ appears which stems from switching of the textures by $J_c$[7]. In the stripe and lattice phases, $\Delta H_1$ increases monotonically with $J_c$ while $\Delta H_2$ decreases. In FMR measurements of a non-textured saturated magnetization, the linewidth is directly related to the magnetic losses, therefore, a current induced linewidth broadening is indicative of a damping-like torque. However, in textured magnets, the relation between the Slonczewski damping-like torque and resonance linewidth is not straight forward due to the

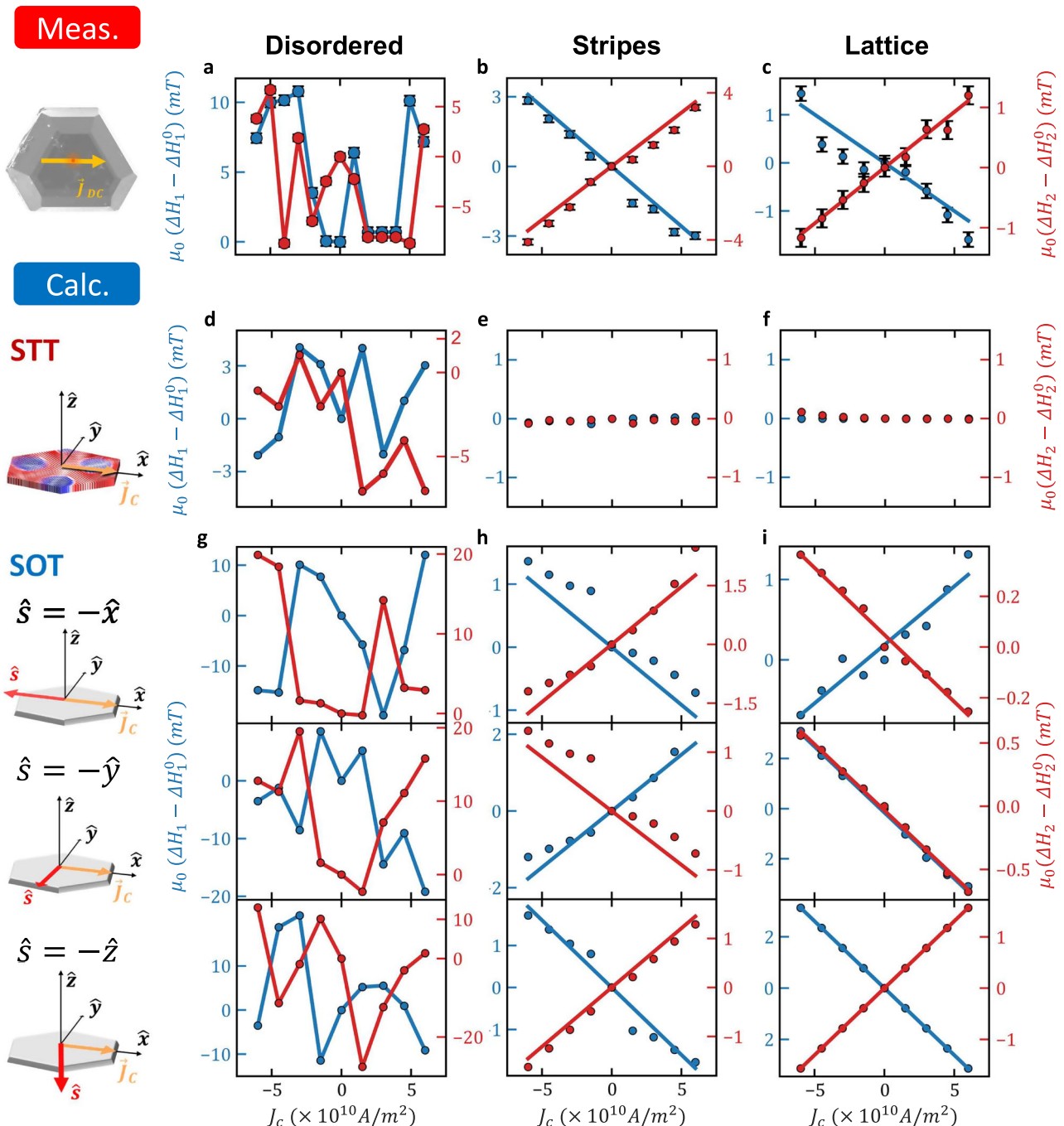

**Fig. 3 | Measured and calculated current induced skyrmion resonance line-width broadening at the disordered, stripe, and bubble lattice phases.**
**a**–**c** Measured $\Delta H_i - \Delta H_i^0$ at 5, 10, and 12 $GHz$. **d**–**f** Calculated $\Delta H_i - \Delta H_i^0$ for DC STT. **g**–**i** Calculated $\Delta H_i - \Delta H_i^0$ for DC SOT. The figures present the calculation for $\hat{s} = -\hat{x}$, $\hat{s} = -\hat{y}$, and $\hat{s} = -\hat{z}$. In the simulations, $\Delta H_i$ for the disordered, stripe, and

lattice phases were calculated at $\mu_0 H_0 = 10$, $100$, and $250\,mT$, respectively, corresponding to the measurements at 5, 10, and 12 $GHz$. Left, middle, and right columns correspond to the disordered, stripe, and bubble lattice phase, respectively. Red and blue solid lines correspond to the CCW and breathing modes, respectively.

anisotropic nature of the torque. To elucidate this behavior, the torque was included in the micromagnetic simulations by introducing STT and SOT terms of the forms $\tau_{STT}\widehat{\boldsymbol{m}}\times\widehat{\boldsymbol{m}}\times(\vec{\boldsymbol{J_c}}\nabla)\widehat{\boldsymbol{m}}$ and $\tau_{SOT}J_c\widehat{\boldsymbol{m}}\times\widehat{\boldsymbol{m}}\times\widehat{\boldsymbol{s}}$, respectively. Here, $\tau_{STT}$ and $\tau_{SOT}$ are coefficients of the two torques and $\hat{s}$ is the spin polarization of the SOT (see 'Methods'). Figure 3d–f present the simulation results for the case of a DC STT by plotting the $J_c$-dependent $\Delta H_i$. In the disordered phase, a $J_c$-induced texture switching takes place which manifests in the random modulation of $\Delta H_i$. However, no discernable modulation of the linewidth is observed

in the stripe and lattice phases. This trend persists up to a critical value of $J_c$ after which the texture switches also in these phases (see Supplementary Note 4). In contrast, the application of a DC SOT leads to the modulation of $\Delta H_i$ observed experimentally. The SOT case was studied for $\hat{s}$ along the three principal axes as presented in Fig. 3g–i. In the disordered phase, the texture switching is reproduced while a monotonic dependence of $\Delta H_i$ on $J_c$ is calculated in the stripe and lattice phases. It is seen that the case of $\hat{s} = -\hat{z}$ best reproduces the measurements. Alternatively, also a combination of $\hat{s} = -\hat{z}$ and $\hat{s} = -\hat{x}$

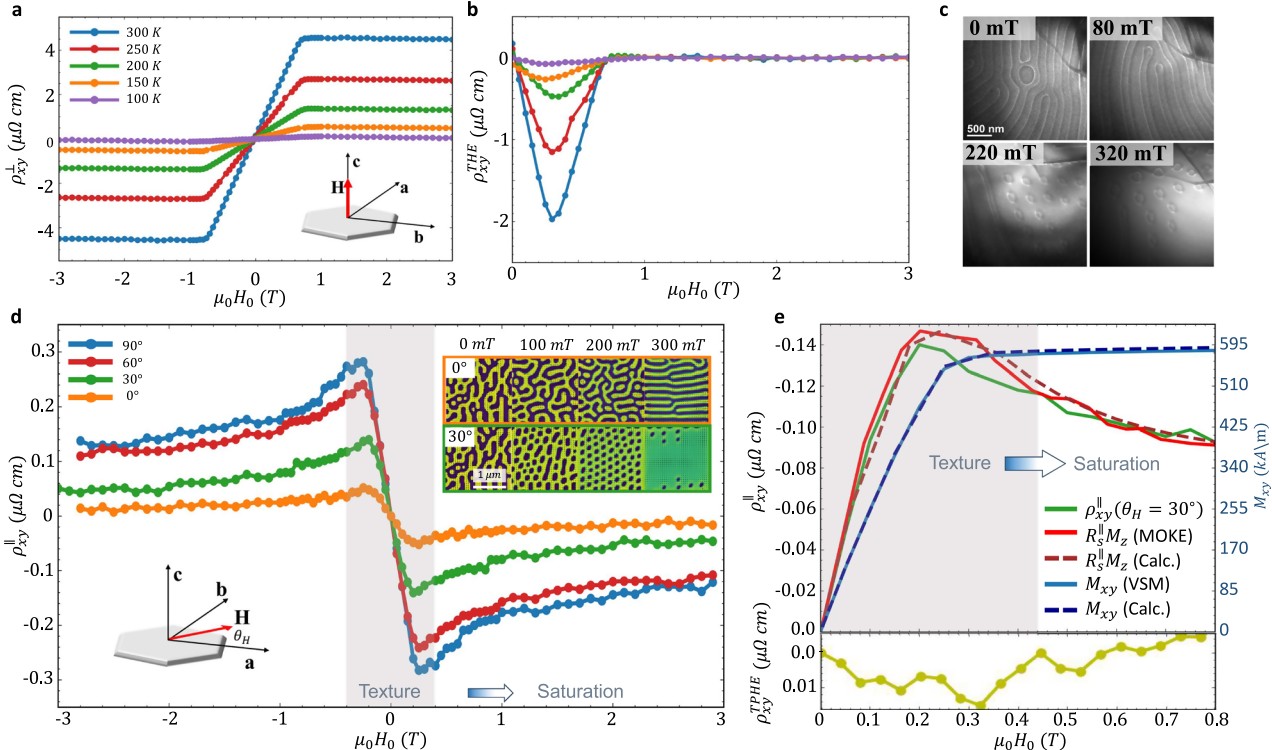

**Fig. 4 | Angle and temperature dependent DC Transport measurements.**
**a** Temperature dependence of $\rho_{xy}^{\perp}$ with OOP $\vec{H}$. **b** Temperature dependence of $\rho_{xy}^{THE}$. $\rho_{xy}^{THE}$ is extracted by subtracting the OHE and AHE contributions from $\rho_{xy}^{\perp}$. **c** LTEM images at 300 K for $\mu_0 H_0 = 0, 80, 220,$ and $320\,mT$ in the OOP configuration. **d** PHE measurements of $\rho_{xy}^{\parallel}$ for IP $\vec{H}$ applied at $\theta_H = 0°, 30°, 60°,$ and $90°$ with respect to the [100] axis. Inset presents calculated magnetic textures for $\theta_H = 0°$ and $30°$. **e** Extraction of the topological contribution, $\rho_{xy}^{TPHE}$, at $\theta_H = 30°$. Upper panel:

Measurement of $\rho_{xy}^{\parallel}$ (green solid line) together with the measured $R_s^{\parallel}\mu_0 M_z$ (red line) at $\theta_H = 30°$. $M_z$ was measured using a MOKE setup at $\theta_H = 30°$ (red line). $R_s^{\parallel}$ was determined from $\rho_{xy}^{\parallel}$ at the saturated magnetization regime. Saturation conditions were verified by calculating $M_z$ at $\theta_H = 30°$ (red dashed line) and by measuring $M_{xy}$ using a VSM (blue solid line). Calculated $M_{xy}$ (blue dashed line) reproduces the measured $M_{xy}$ in saturation. Lower panel presents the extracted $\rho_{xy}^{TPHE}$.

may result in a similar trend. In the stripe phase, the inhomogeneous broadening is more significant. Therefore, the effective spin Hall angle was estimated from the CCW mode at the lattice phase for $\hat{s} = -\hat{z}$ which resulted in $\theta_{SH}^{eff} = 0.027 \pm 0.006$ (see 'Methods').

The THE is a useful probe of texture-driven transport phenomena and appears as an anomaly in the AHE response[38,39]. Figure 4a presents temperature dependent Hall resistivity $\rho_{xy}^{\perp}$ which consists of the contributions of the ordinary Hall effect (OHE), AHE, and THE[40]. For high $|H_0|$, $M_z$ is saturated so that the THE vanishes and the AHE dominates. At lower $|H_0|$, the THE arises. This is readily seen in Fig. 4b by plotting the THE contribution, $\rho_{xy}^{THE}$, obtained by subtracting the contributions of the OHE and AHE (see 'Methods'). $\rho_{xy}^{THE}$ survives up to 700 mT where the maximal value occurs at $\mu_0 H_0 \approx 300\,mT$ and accounts for nearly 50% of $\rho_{xy}^{\perp}$. This highlights the significant role of the real-space Berry curvature in $Fe_3Sn_2$[9]. Lorentz transmission electron microscopy (LTEM) images presented in Fig. 4c further support this conclusion revealing that a skyrmion lattice forms having $Q \neq 0$ at the maximal value of $\rho_{xy}^{THE}$.

In contrast to the measurement of the THE, in the OSTFMR measurements $\vec{H}$ was applied in the sample plane. Therefore, the planar Hall resistivity $\rho_{xy}^{\parallel}$ was also measured and is presented in Fig. 4d. In this geometry, the tilted UMA produces a non-vanishing $M_z$ giving rise to the usual AHE that originates in the $k$-space band topology in addition to the effective magnetic field that emerges from the localized spin structure in real-space. We refer to these contributions as the $k$-PHE and topological PHE (TPHE) analogously to the AHE and THE terms of $\rho_{xy}^{\perp}$. Accordingly, $\rho_{xy}^{\parallel} = R_s^{\parallel}\mu_0 M_z + P_z^{\parallel}(H_0)R_0^{\parallel}b^z$ where $R_s^{\parallel}$ and $R_0^{\parallel}$ are the $k$-PHE and TPHE coefficients, respectively, $P_z^{\parallel}(H_0)$ is the net $\hat{z}$ spin polarization of the charge carriers, and $b^z$ is the fictitious gauge field emerging from

the real-space Berry curvature. $P_z^{\parallel}(H_0) = C_0 \cdot |M_z/M_s|$ with the constant $C_0$ determined by the material band structure and $M_s$ is the saturation magnetization. The figure presents $\rho_{xy}^{\parallel}$ for different IP $\theta_H$ angles together with the calculated magnetic textures at $\theta_H = 0°$ and $30°$ (textures for $\theta_H = 60°, 90°$ are presented in Supplementary Note 5). $\rho_{xy}^{\parallel}$ is substantially smaller than $\rho_{xy}^{\perp}$ due to the smaller $M_z$ and $P_z^{\parallel}$ in the planar geometry. The non-monotonic behavior of $\rho_{xy}^{\parallel}$ is a hallmark of the formation of a multi-domain texture in PHE measurements[41]. For $\theta_H = 30°$ at which the OSTFMR measurements were carried out, the maximal value of $\rho_{xy}^{\parallel}$ occurs at 198 mT corresponding to the bubble lattice phase. At $\theta_H = 0°$, the simulations reveal that the lattice phase does not form, therefore, $\rho_{xy}^{\parallel}$ should not exhibit a peak. Indeed, the measured $\rho_{xy}^{\parallel}$ is smaller as compared to the case of $\theta_H = 30°$, however, a slight peak is still observed and stems from a slight misalignment of $\vec{H}$ as verified in Supplementary Note 6. At $\theta_H = 30°$ and $|\mu_0 H_0| > 400\,mT$, the texture saturates resulting in $b^z \approx 0$ and the response purely stems from the $k$-space PHE. In the presence of the tilted UMA axis, $M_z$ saturates to different values depending on $\theta_H$. Consequently, $\rho_{xy}^{\parallel}$ approaches a different limit as $\theta_H$ is varied. The different saturation limits were also verified in the calculation (see Supplementary Note 6).

To determine the TPHE contribution at $\theta_H = 30°$, $M_z$ was measured using a static MOKE. This data is presented in Fig. 4e (red solid line) together with the measured $\rho_{xy}^{\parallel}(\theta_H = 30°)$ (green solid line). $M_z$ is represented by plotting $R_s^{\parallel}\mu_0 M_z$ where $R_s^{\parallel}$ was determined from the saturated $\rho_{xy}^{\parallel}$ responses at high $H_0$. The TPHE contribution $\rho_{xy}^{TPHE}$ was extracted by subtracting $R_s^{\parallel}\mu_0 M_z$ from $\rho_{xy}^{\parallel}$ and is plotted in the lower panel of the figure. This procedure was additionally validated by vibrating sample magnetometer (VSM) measurements of the IP magnetization, $M_{xy}$ (blue solid line), and a numerical calculation of $M_z$ and

$M_{xy}$ (red and blue dashed lines, respectively) which reproduced the measurement in the saturation regime. It is seen that $\rho_{xy}^{\parallel}$ primarily stems from the $k$-space PHE while the maximal contribution of $\rho_{xy}^{TPHE}$ is $\sim 10.7\%$ obtained at $\sim 320\,mT$ and is significantly smaller than the relative contribution of the THE in the measurement of $\rho_{xy}^{\perp}$. Once more, the maximal value of $\rho_{xy}^{TPHE}$ corresponds to a lattice phase as seen from the calculated textures displayed in Fig. 4d. At $250\,mT$ where $\theta_{SH}^{eff}$ was extracted, $\rho_{xy}^{TPHE}$ accounts for $8.5\%$ of $\rho_{xy}^{\parallel}$.

## Discussion

The PHE transport measurements illustrate that the spin texture plays a small yet finite role in the carrier transport mechanisms. These results suggest a possible contribution of the magnetic texture also to the spin transport. While our simulations imply a negligible effect of the DC STT on the resonance linewidth, the SHE or interfacial spin-orbit interactions may certainly be the dominant mechanisms. The predictions of the possible THE contribution due to the real-space magnetic topology of the localized spin structures is relevant in the adiabatic regime[21–24]. In this regime, the carrier and the scatterer are strongly coupled such that the itinerant electrons follow the direction of the local magnetization and the spin-dependent emergent magnetic field due to the real-space Berry curvature arises[24]. The coupling strength is given by the adiabatic parameter $\lambda = \tau_{fl}/\tau_{ex}$ which represents the ratio between the time of flight through the localized spin texture and the exchange-driven spin-flip time[42]. The exchange splitting energy was estimated to be $\sim 22\,meV$ (see 'Methods'), in agreement with recent measurements[43]. Accordingly, $\tau_{ex} \sim 30\,fs$ was determined. For the spin texture diameter of $100\,nm$ and Fermi velocity $v_F \approx 10^5 - 10^6\,ms^{-1}$ [12,44], $\tau_{fl}$ was estimated to be $10^{-12} - 10^{-13}\,s$. Therefore, $\lambda \approx 3 - 30$ indicating an intermediate coupling regime in which the transition from spin-dependent to spin-independent scattering occurs[42]. For the extreme case in which all scattering mechanisms are spin-dependent, the texture dependent and texture independent spin Hall conductivities are $\sim 726 \pm 161\,(\hbar/e)\,(\Omega cm)^{-1}$ and $\sim 67 \pm 15\,(\hbar/e)\,(\Omega cm)^{-1}$, respectively.

In this work, we studied the skyrmion resonance response in a frustrated $Fe_3Sn_2$ magnet. The dynamics were excited by a self-induced RF spin-torque rather than an RF Oersted field, while the magnetization response was probed optically in a time-resolved measurement. Our results indicated that only the CCW and breathing modes are excited, confirming the theoretical predictions for frustrated magnets. Although the spin-torque is anisotropic, our experiments revealed that the application of a DC bias current is capable of modulating the resonance linewidth in a fashion that is reminiscent of conventional SHE measurements in which an adjacent FM layer is used as a spin-current sensor. Micromagnetic simulations illustrated that the skyrmion resonance linewidth is sensitive to an SOT-like mechanism rather than a DC STT. In $Fe_3Sn_2$, for the given planar geometry, transport measurements suggest a primary $k$-space contribution to an SOT-type mechanism and a smaller contribution of the real-space spin texture. This is another indication of the generally weaker contribution of the real-space spin texture as compared to the $k$-space band topology.

These results provide insights into the anisotropic nature of spin-torque in frustrated magnets and advances the possibility of using the skyrmion resonance as a sensor for spin currents. Natural extensions of the work include the search for more pronounced real-space effects anticipated in the strongly adiabatic regime for a topologically non-trivial scatterer. To gain a deeper understanding of the interplay between real-space and momentum-space induced phenomena, it would be valuable to explore a variety of magnetic textures, including trivial and topologically protected states of higher $Q$. Relevant materials include helimagnets, ferromagnets, additional frustrated magnets, and antiferromagnets. These encompass systems with bulk DMI, interfacial DMI, strong uniaxial magnetic anisotropies (UMA), and defect-engineered structures. Of particular interest are thin film systems that are pivotal to the future applications. A more comprehensive study will contribute in identifying the universal laws that govern the skyrmion resonance dynamics[29] in the presence of spin polarized currents. Having the ability to address the skyrmion dynamics using optical techniques, rather than relying on microwave absorption, provides advantages in sensitivity and offers a more direct, artifact-free approach. The applicability of our approach is expected to have broad ramifications for the control of magnetic textures in a variety of applications including data storage and transfer, neuromorphic and reservoir computing, and magnetic sensing, with advantages in density, power, and scalability.

## Methods

### Crystal growth

The $Fe_3Sn_2$ polycrystalline powder and $I_2$ (mass ratio $10:1$) were sealed in vacuum quartz tube which was then placed in a horizontal three-temperature-zone tube furnace with a temperature gradient from $650\,°C$ to $750\,°C$. The raw material was kept at $650\,°C$ for $\sim 150$ hours and then quenched into cold water. The resultant crystal dimensions were roughly $\sim 0.85 \times 1\,mm^2$ and $55\,\mu m$ thick having resistivity of $\rho_{xx} = 68.1\,\mu\Omega \cdot cm$. X-ray diffraction data revealed a hexagonal surface in the (001) plane (see Supplementary Note 1).

### OSTFMR technique

The dynamics were probed using a Ti:Sapphire laser emitting $35\,fs$ pulses at $800\,nm$ and a repetition rate of $80\,MHz$. The laser was focused to a probing spot size of $\sim 10\,\mu m$. $J_c$ was estimated from the Poisson equation while accounting for the geometrical dimensions of the crystal (see Supplementary Note 2).

### Micromagnetic simulations

Micromagnetic simulations were performed using object oriented micromagnetic framework (OOMMF)[45] taking the energy density: $E_T = -A\vec{m} \cdot \nabla^2 \vec{m} - \frac{1}{2}\mu_0 M_s \vec{m} \cdot \vec{H}_d - K(\vec{m} \cdot \hat{u})^2 - \mu_0 M_s \vec{m} \cdot \vec{H}$, where $M_s = 566\,kA/m$, and the exchange stiffness and anisotropy constants were $A = 2 \cdot 10^{-11}\,J/m$ and $K = 1.8 \cdot 10^5\,J/m^3$, respectively. $\hat{u}$ is the unit vector of the anisotropy axis which was tilted at $22°$ from the OOP axis along $\hat{x}$[5,9]. $\vec{H}_d$ is the demagnetization field and was evaluated numerically. The exchange length under these parameters is $\sqrt{A/0.5\mu_0 M_s^2} = 9.97\,nm$. The calculation space was $2 \times 2\,\mu m^2$ and $100\,nm$ thick and was discretized into $5 \times 5 \times 5\,nm^3$ unit cells. The initial state was determined by initializing each unit cell to a random value after which the magnetization relaxed to its ground state. The frequency response was obtained by taking the Fourier transform of the calculated impulse response at a constant $H_0$.

### STT modeling

The STT was incorporated into the micromagnetic simulations by including the Zhang-Li term[46], $-\frac{1+\alpha\beta}{1+\alpha^2}\hat{m} \times (\hat{m} \times (\vec{u} \cdot \nabla)\hat{m})$, where the non-adiabatic coefficient $\beta$ was set to 0.1 as in ref. [47]. The Zhang-Li coefficient is given by $\vec{u} = \frac{J_c P \mu_B}{e M_s}$ for which $\mu_B$ is the Bohr magneton, $e$ is the elementary charge. The polarization rate was set to $P = 1$ to estimate the upper limit of the STT. The linewidth was calculated from the Fourier transform of the impulse response.

### SOT modeling and extraction of $\theta_{SH}^{eff}$

The generation of spin currents within the calculated volume was modeled assuming an average spin-flip length of $\lambda_{flip}$ in which the spin polarized electrons transfer their spin angular momentum. This torque was modeled by introducing the Slonczewski spin-torque $-\gamma\frac{1}{\mu_0 M_s}\frac{\hbar}{2e}\frac{J_c}{\lambda_{flip}}\theta_{SH}^{eff}(\hat{m} \times \hat{m} \times \hat{s})$ which was injected into each discretized

cell. Here, $\theta_{SH}^{eff} = J_s/J_c$ with $J_s$ being the spin current density and $\hbar$ is the reduced Planck's constant. $\lambda_{flip}$ determined the cell size. The inter-cell propagation of spin angular momentum was set to zero. A $\lambda_{flip}$ of $5\,nm$ was estimated from the Fermi velocity and the exchange driven spin flip time by $v_F \cdot \tau_{ex}$ using typical values in FMs[48].

## Lorentz TEM
The LTEM measurements were performed by Tecnai F20 in the Lorentz mode. The magnetic contrast of the skyrmions was determined from a set of three images with under-, over-, and in-focus. Samples were prepared by using focused ion beam (FIB) and the corresponding orientation was determined by selected-area electron diffraction (SAED).

## Electrical transport and magnetization measurements
The magnetization and transport properties were determined using a physical property measuring system (PPMS, Quantum Design). The electrical properties were measured by standard six-probe method with $J_c$ flowing in the [100] direction. The Hall resistivity is given by $\rho_{xy}^{\perp} = R_0^{\perp}\mu_0 H_0 + R_s^{\perp}\mu_0 M_z + P_z^{\perp}(H_0)R_0^{\perp}b^z$. $R_0^{\perp}$ and $R_s^{\perp}$ are the OHE and AHE coefficients, respectively. The third term is the THE resistivity $\rho_{xy}^{THE}$ in which $P_z^{\perp}(H_0)$ is the net $\hat{z}$ spin polarization of the charge carriers for an OOP $\vec{H}$. $M_z$ was measured using a VSM.

## Reporting summary
Further information on research design is available in the Nature Portfolio Reporting Summary linked to this article.

## Data availability
The data that support the findings of this study are available from the corresponding author upon request.

## Code availability
The codes used in theoretical calculations are available from the corresponding author upon request.

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

## Acknowledgements

This work was supported by the Israel Science Foundation and the National Natural Science Foundation of China (Grant No. 3011/23 and Grant No. 12361141823), the Synergetic Extreme Condition User Facility (SECUF), the Israel Science Foundation (Grant No. 1217/21), and the Cooperation in Science and Technology action (CA23136 CHIROMAG). N.B., B.A., and A.C. thank the Harvey M. Krueger family center of Nanoscience and Nanotechnology at the Hebrew University of Jerusalem and the Lipper and Peter Brojde foundations.

## Author contributions

N.B., W.W., and A.C. conceived the research. Y.C.L., W.W., I.R., and A.C. supervised the experiments. H.L., Y.C.L., and W.W. fabricated and characterized the sample. N.B. performed the dynamical measurements. N.B. and B.A. performed the simulations. H.L. performed the transport measurements and the LTEM imaging. All authors discussed the results and co-wrote the manuscript. A.C. and W.W. directed the research.

## Competing interests

The authors declare no competing interests.
