## [Peer Review File · Nature Communications]

Spin-torque skyrmion resonance in a frustrated magnet

Corresponding Author: Professor Amir Capua

Version 0:

Reviewer comments:

Reviewer #1

(Remarks to the Author)

In this paper, the authors experimentally observe the resonance modes of the skyrmion phase at microwave frequencies in a skyrmion-hosting material due to magnetic frustration, Fe_3Sn_2 . The technique used in the experiments is the Phase Resolved Optically Probed FMR technique. The method directly detects the magnetic resonance modes excited by the radio-frequency AC currents as time profiles of the out-of-plane magnetization by means of Kerr-rotation measurements with synchronized laser pulses. I cannot recommend publication of this paper in Nature Communications in the present form but may recommend it if the authors address my concerns and make major revisions on the manuscript. I'd like to explain my concerns on this paper below.

The results of this paper have two important implications,

1. the direct experimental confirmation that the skyrmion crystal phase in frustrated magnets have only two magnetic resonance modes, called the breathing mode and the counterclockwise rotation mode. It is known that skyrmion crystals stabilized by the DM interactions in chiral magnets and polar magnets have three modes, that is, the above two modes and a clockwise rotation mode. On the other hand, it has been theoretically predicted that skyrmion crystals stabilized by magnetic frustrations instead of the DM interactions do not have the clockwise rotation mode. The present results are the first experimental demonstration of this theoretical prediction to my knowledge. Magnetic excitations in magnetic materials are important elementary excitations that are origins of various physical phenomena and material functions. The experimental results, which revealed their properties, are expected to provide an important starting point for future research into the rich physics of frustrated magnetic skyrmions and other topological magnetisms.

2. The successful demonstration of the new experimental technique to observe magnetic resonance modes, i.e., the Phase Resolved Optically Probed FMR, is also a significant achievement of this work. To observe the resonance modes of skyrmions and other topological magnetisms, the microwave absorption experiments based on the radio-frequency measurements are usually used. On the other hand, the present experimental technique based on the Kerr-effect measurements is a relatively new experimental technique. In the present study, the magnetic excitation modes in different magnetic phases of Fe_3Sn_2 have been clearly observed, which successfully demonstrates the efficiency and usefulness of this new experimental technique.

This paper thus contains significant achievements, but its significance and importance are not very well explained in the introductory part. Worst of all, there is an obvious misunderstanding of the spin Hall effect and the topological Hall effect. The authors use the term topological spin Hall effect (TSHE) many times in this paper, but as far as I know, it is the "topological Hall effect" but not the "topological spin Hall effect" that is manifested by skyrmions through the real-space spin chirality. I would like to discuss this point further below.

First, it is not clear what the authors mean when they claim that the spin current can be detected by measuring the magnetic resonance excitations using this experimental method. The origin of the magnetic resonance excitations is the oscillation of the skyrmions driven by the AC electric current, but the basis for the claim that this is a spin current or a spin current generated by the topological spin Hall effect due to the skyrmions, is unclear. Current-driven dynamics of skyrmions in bulk materials is usually through spin-transfer torques. In this case, the skyrmions are driven by spin-polarized electric currents rather than spin currents. In addition, it is certainly possible for Fe_2Sn_3 as a layered material to generate a vertical spin current due to the spin Hall effect caused by interfacial spin-orbit interactions. This vertical spin current could possibly drive the skyrmions by exerting the spin-orbit torque. However, this spin current is a vertical spin current originating from the

interface spin-orbit interaction, which is completely different from the spin current originating from the "topological spin Hall effect" caused by skyrmions.

More to the point, the term and phenomenon of the "topological spin Hall effect" seem to be not a widely accepted concept. The phenomenon caused by the emergent magnetic field due to the magnetic topology of skyrmions is the topological Hall effect, and I have never heard a phenomenon called the topological spin Hall effect. I think that most researchers have never heard that a phenomenon called the topological spin Hall effect which is illustrated in Fig.1a really occurs.

In view of the above, the authors' logic that the spin current can be detected by the present experimental method via measuring the skyrmion resonance modes driven by spin currents generated by the skyrmion-induced topological spin Hall effect is rather implausible. Nevertheless, the introduction of the present paper is tailored based on this discussion. And there seem to be some rather fatal errors and confusions in this introduction. Let me briefly summarize the widely accepted understanding of the spin Hall effect and the topological Hall effect.

First, the spin Hall effect proposed by Murakami and Nagaosa in Science journal, which the authors call iSHE, is indeed caused by a topology in the momentum space, but it is a topology of the band structure and has nothing to do with the magnetic topology of the localized spin structure or the spin chirality.

Second, as discussed in PRB 62, R6065 (2000) and Science 291, 2573 (2001), the band topology or the Berry phase in the momentum space produced by the spin chirality of the localized spin structure causes the topological Hall effect, but it is not the spin Hall effect.

Third, it is well known that the real-space spin chirality or the real-space magnetic topology of localized spin structures like skyrmions, can cause the topological Hall effect (but not the topological spin Hall effect).

Fourth, it is also known that there are both the contribution from the real-space topology and that from the momentum-space topology to the topological Hall effect caused by the topological magnetic structures of the localized spins like skyrmions.

On the contrary, it is not widely accepted that a phenomenon called the topological spin Hall effect is caused by skyrmions. Moreover, if such a phenomenon really occurs, it is not obvious at all how the spin currents generated via this phenomenon would be related to the magnetic resonance excitations observed experimentally in the present study. The explanation required for such a claim is missing in the paper.

I believe that the significance of this paper lies in the two points mentioned above and that they are important results worth publication in Nature Communications. However, the introduction part of this paper does not mention the significance of this paper. On the other hand, the authors' claim that the spin currents generated via the skyrmion-induced topological spin Hall effect can be detected via the skyrmion resonance measurements contains some errors and is not supported by firm argument. We think that active assertion of this claim is a major detriment to the value of this paper. In conclusion, we believe that if the entire paper including the introduction part is revised to correctly discuss the results and significance of this research, we may recommend this paper for publication in Nature Communications.

Reviewer #2

(Remarks to the Author)

The application of skyrmion resonance as a sensor for spin currents via the OSTFMR technique is innovative. Although spin-torque resonance methods are established, using them to probe skyrmion dynamics in a frustrated magnet like Fe_3Sn_2 is creative and may yield fresh insights into spin-torque effects in complex magnetic textures. By examining both the intrinsic spin Hall effect (iSHE) and the topological spin Hall effect (TSHE) in the same material, this study provides valuable insights into a system where both k-space and real-space chirality contribute to spin transport. The finding that iSHE is dominant while TSHE plays a minor role advances our understanding of spin currents in nontrivial magnetic textures and the interplay between iSHE and TSHE in frustrated systems.

The authors present consistent experimental results and micromagnetic simulations, offering a detailed analysis of skyrmion resonance modes and current-induced effects. Their quantitative extraction of parameters, such as effective spin Hall conductivity and Gilbert damping, enhances the rigor of the manuscript. Fe_3Sn_2 is a relatively rare example of a frustrated magnet with robust skyrmion stability at high temperatures and a unique independence from spin-orbit coupling (due to frustration rather than Dzyaloshinskii-Moriya interaction), distinguishing it from more conventional skyrmion systems. This work advances previous studies by investigating linewidth modulation under both AC and DC currents and exploring the anisotropic nature of spin-torque in frustrated systems.

While this study may not be a groundbreaking contribution to skyrmion research overall, as skyrmion dynamics and spin-torque effects are well-established, it adds meaningful insights specific to frustrated magnets and spin current detection. To broaden its impact, the authors could suggest potential applications or implementations of this technique, as well as possible extensions to other materials and practical devices.

The research is technically sound, innovative, and well-suited for Nature Communications, though it may not be considered transformative. Its strengths lie in the novel application of OSTFMR to skyrmion resonance and the detailed analysis of spin Hall effects in Fe_3Sn_2 .

Other suggestions:

Line 295: Change "100" to "[100]."

Figure 2 Caption: Change "Each response was normalized to the peak value." to "Each response was normalized to its peak value."

Abstract: Add error bars to the effective spin Hall conductivity for precision.

In the abstract and conclusion, it would be beneficial to emphasize the unique contributions of this research, such as insights into the anisotropic nature of spin-torque in frustrated magnets and advancements in skyrmion resonance as a sensor for spin currents.

Version 1:

Reviewer comments:

Reviewer #1

(Remarks to the Author)

After careful reading of the authors' replies and the revised manuscript, I found that the authors have carefully addressed my questions, suggestions, and criticisms raised in my first report and have revised the manuscript appropriately. In the authors' reply letter, the main points of my comments are summarized as follows. These summaries indicated that the authors completely and correctly understood my comments,

The reviewer acknowledges the novelty of our work but notes that it is not well represented in the manuscript, particularly in the introduction.

The role of spin-transfer torque (STT) was not considered.

The introduction contains some errors in the descriptions of the spin Hall effect (SHE) and topological Hall effect (THE).

The topological SHE (TSHE) seems to be not a widely accepted phenomenon.

The connection between the TSHE and the measured dynamics is not sufficiently well established in the manuscript.

The reviewer recommends removing the active assertion of the TSHE.

In my previous report, I stated that the results of this paper are very significant in two respects and worthy of publication in Nature Communications. On the other hand, as my concerns, I pointed out that this importance and background were not properly discussed in the introduction and that several terms were used inappropriately or even incorrectly. In the revised manuscript, I found that the introduction has been completely rewritten so that the importance of the results can be understood by broad readers. I also found that the terminology of the terms has been clarified, and the revised manuscript uses the terms correctly. Furthermore, the title has been changed. I think this new title is good because it expresses the essential importance of the research results in correct terminology. In light of the above, I now recommend publication of this revised manuscript in Nature Communications.

Reviewer #2

(Remarks to the Author)

The authors have addressed my comments, and I recommend the manuscript be published in Nature Communications.

Response to comments of Reviewer #1

We appreciate the reviewer's effort in evaluating our work. For their convenience, we include their full report below and address their comments point by point afterwards.

Reviewer #1 (Remarks to the Author):

"In this paper, the authors experimentally observe the resonance modes of the skyrmion phase at microwave frequencies in a skyrmion-hosting material due to magnetic frustration, Fe_3Sn_2 . The technique used in the experiments is the Phase Resolved Optically Probed FMR technique. The method directly detects the magnetic resonance modes excited by the radio-frequency AC currents as time profiles of the out-of-plane magnetization by means of Kerr-rotation measurements with synchronized laser pulses. I cannot recommend publication of this paper in Nature Communications in the present form but may recommend it if the authors address my concerns and make major revisions on the manuscript. I'd like to explain my concerns on this paper below.

The results of this paper have two important implications.

1. the direct experimental confirmation that the skyrmion crystal phase in frustrated magnets have only two magnetic resonance modes, called the breathing mode and the counterclockwise rotation mode. It is known that skyrmion crystals stabilized by the DM interactions in chiral magnets and polar magnets have three modes, that is, the above two modes and a clockwise rotation mode. On the other hand, it has been theoretically predicted that skyrmion crystals stabilized by magnetic frustrations instead of the DM interactions do not have the clockwise rotation mode. The present results are the first experimental demonstration of this theoretical prediction to my knowledge. Magnetic excitations in magnetic materials are important elementary excitations that are origins of various physical phenomena and material functions. The experimental results, which revealed their properties, are expected to provide an important starting point for future research into the rich physics of frustrated magnetic skyrmions and other topological magnetisms.

2. The successful demonstration of the new experimental technique to observe magnetic resonance modes, i.e., the Phase Resolved Optically Probed FMR, is also a significant achievement of this work. To observe the resonance modes of skyrmions and other topological magnetisms, the microwave absorption experiments based on the radio-frequency measurements are usually used. On the other hand, the present experimental technique based on the Kerr-effect measurements is a relatively new experimental technique. In the present study, the magnetic excitation modes in different magnetic phases of Fe_3Sn_2 have been clearly observed, which successfully demonstrates the efficiency and usefulness of this new experimental technique.

This paper thus contains significant achievements, but its significance and importance are not very well explained in the introductory part. Worst of all, there is an obvious misunderstanding of the spin Hall effect and the topological Hall effect. The authors use the term topological spin Hall effect (TSHE) many times in this paper, but as far as I know, it is the "topological Hall effect" but not the "topological spin Hall effect" that is manifested by skyrmions through the real-space spin chirality. I would like to discuss this point further below.

First, it is not clear what the authors mean when they claim that the spin current can be detected by measuring the magnetic resonance excitations using this experimental method. The origin of the magnetic resonance excitations is the oscillation of the skyrmions driven by the AC electric current, but the basis for the claim that this is a spin current or a spin current generated by the topological spin Hall effect due to the skyrmions, is unclear. Current-driven dynamics of skyrmions in bulk materials is usually through spin-transfer torques. In this case, the skyrmions are driven by spin-polarized electric currents rather than spin currents. In addition, it is certainly possible for Fe_2Sn_3 as a layered material to generate a vertical spin current due to the spin Hall effect caused by interfacial spin-orbit interactions. This vertical spin current could possibly drive the skyrmions by exerting the spin-orbit torque. However, this spin current is a vertical spin current originating from the interface spin-orbit interaction, which is completely different from the spin current originating from the "topological spin Hall effect" caused by skyrmions.

More to the point, the term and phenomenon of the "topological spin Hall effect" seem to be not a widely accepted concept. The phenomenon caused by the emergent magnetic field due to the magnetic topology of skyrmions is the topological Hall effect, and I have never heard a phenomenon called the topological spin Hall effect. I think that most researchers have never heard that a phenomenon called the topological spin Hall effect which is illustrated in Fig.1a really occurs.

In view of the above, the authors' logic that the spin current can be detected by the present experimental method via measuring the skyrmion resonance modes driven by spin currents generated by the skyrmion-induced topological spin Hall effect is rather implausible. Nevertheless, the introduction of the present paper is tailored based on this discussion. And there seem to be some rather fatal errors and confusions in this introduction. Let me briefly summarize the widely accepted understanding of the spin Hall effect and the topological Hall effect.

First, the spin Hall effect proposed by Murakami and Nagaosa in Science journal, which the authors call iSHE, is indeed caused by a topology in the momentum space, but it is a topology of the band structure and has nothing to do with the magnetic topology of the localized spin structure or the spin chirality.

Second, as discussed in PRB 62, R6065 (2000) and Science 291, 2573 (2001), the band topology or the Berry phase in the momentum space produced by the spin chirality of the localized spin structure causes the topological Hall effect, but it is not the spin Hall effect.

Third, it is well known that the real-space spin chirality or the real-space magnetic topology of localized spin structures like skyrmions, can cause the topological Hall effect (but not the topological spin Hall effect).

Fourth, it is also known that there are both the contribution from the real-space topology and that from the momentum-space topology to the topological Hall effect caused by the topological magnetic structures of the localized spins like skyrmions.

On the contrary, it is not widely accepted that a phenomenon called the topological spin Hall effect is caused by skyrmions. Moreover, if such a phenomenon really occurs, it is not obvious at all how the spin currents generated via this phenomenon would be related to the magnetic resonance excitations observed experimentally in the present study. The explanation required for such a claim is missing in the paper.

I believe that the significance of this paper lies in the two points mentioned above and that they are important results worth publication in Nature Communications. However, the introduction part of this paper does not mention the significance of this paper. On the other hand, the authors' claim that the spin currents generated via the skyrmion-induced topological spin Hall effect can be detected via the skyrmion resonance measurements contains some errors and is not supported by firm argument. We think that active assertion of this claim is a major detriment to the value of this paper. In conclusion, we believe that if the entire paper including the introduction part is revised to correctly discuss the results and significance of this research, we may recommend this paper for publication in Nature Communications."

Point by point response:

General comment:

We thank the reviewer for thoroughly reading our manuscript and for their detailed response. Reading carefully their remarks, we can summarize their main concerns as follows:

- The reviewer acknowledges the novelty of our work but notes that it is not well represented in the manuscript, particularly in the introduction.
- The role of spin-transfer torque (STT) was not considered.
- The introduction contains some errors in the descriptions of the spin Hall effect (SHE) and topological Hall effect (THE).
- The topological SHE (TSHE) seems to be not a widely accepted phenomenon.
- The connection between the TSHE and the measured dynamics is not sufficiently well established in the manuscript.
- The reviewer recommends removing the active assertion of the TSHE.

In response to the reviewer's report, the manuscript was thoroughly revised. The amendments include the following:

- Rewriting the abstract and the entire introduction to express the two implications:
 - 1) The measurement of skyrmion resonance dynamics in frustrated magnets and confirmation of the quenching of the clockwise mode.
 - 2) Demonstration of the time-resolved optically-probed spin-torque skyrmion resonance as a new experimental technique.
- Adoption of a more accurate terminology for the SHE and THE.
- Inclusion of a new study of the influence of STT showing that STT cannot account for the observed current induced linewidth broadening in contrast to a spin-orbit torque (SOT). This remedy includes updating figures and adding a new supplemental materials section. Additionally, a clear distinction between the two types of the spin-torque: STT and SOT, is made throughout the manuscript.
- Removal of substantial sections describing the TSHE and omitting the term TSHE. Likewise, the related schematic figure and the active assertion related to the TSHE were removed.
- Clearly separating between the factual experimental observations and their interpretation.
- As a result of the above modifications, the title was changed and reads now: “Spin-torque skyrmion resonance in a frustrated magnet” where the phrase “induced by spin-polarized currents” was omitted.

Below we elaborate in detail on each of the points:

Point 1.

“...The results of this paper have two important implications, ...

1. The direct experimental confirmation that the skyrmion crystal phase in frustrated magnets have only two magnetic resonance modes...
2. The successful demonstration of the new experimental technique ...

This paper thus contains significant achievements, but its significance and importance are not very well explained in the introductory part.”

Answer: The introduction has been rewritten in its entirety. The discussion presenting the work in light of the interplay between the TSHE and intrinsic SHE (iSHE) has been completely removed. Instead, the new introduction focuses on the study of the skyrmion resonance dynamics in frustrated magnetic textures. Specifically, on the observation of the two breathing and rotational counterclockwise resonance modes rather than the three modes typically found in DMI-dominated magnetic textures. Additionally, the introduction focuses on the new experimental technique as compared to the previously demonstrated microwave absorption methods.

Accordingly, also the abstract and summary were thoroughly revised and updated.

Point 2

“... there is an obvious misunderstanding of the spin Hall effect and the topological Hall effect...”

First, the spin Hall effect proposed by Murakami and Nagaosa in Science journal, which the authors call iSHE, is indeed caused by a topology in the momentum space, but it is a topology of the band structure and has nothing to do with the magnetic topology of the localized spin structure or the spin chirality.

Second, as discussed in PRB 62, R6065 (2000) and Science 291, 2573 (2001), the band topology or the Berry phase in the momentum space produced by the spin chirality of the localized spin structure causes the topological Hall effect, but it is not the spin Hall effect.

Third, it is well known that the real-space spin chirality or the real-space magnetic topology of localized spin structures like skyrmions, can cause the topological Hall effect (but not the topological spin Hall effect).

Fourth, it is also known that there are both the contribution from the real-space topology and that from the momentum-space topology to the topological Hall effect caused by the topological magnetic structures of the localized spins like skyrmions.”

Answer: The description of the SHE and THE has been revised and the terminology related to spin currents, spin polarized electric currents, spin-torque, SOT, STT, and spin chirality throughout the manuscript was revisited. The following modifications were made:

- A new paragraph discussing the SHE and THE was added to the introduction. This section now provides a clearer and more comprehensive explanation of the various mechanisms that may generate spin torques in Fe_3Sn_2 . The k -space contribution to the THE was lacking in the original manuscript and is now explained as well following Refs. [Ohgushi et al. PRB **62**, R6065 (2000) and Taguchi et al. Science **291**, 2573 (2001)]. Likewise, the contribution of the interfacial SOC to the SHE was included.
- Ambiguous and less precise terms describing the spin-torques, particularly those that did not distinguish between STT and SOT, have been removed. STT now refers exclusively to the torque emerging from spin-polarized electric currents rather than spin current. SOT refers to the torque emerging from spin currents. The broader term ‘spin-torque’ is now used to refer to both STT and SOT. Where required, terms like ‘spin current’ were replaced by the more accurate ‘spin-torque’ term.
- A new figure that illustrates schematically the contributions to the SHE and THE instead of Fig. 1(a).

Point 3

“The authors use the term topological spin Hall effect (TSHE) many times in this paper, but as far as I know, it is the ‘‘topological Hall effect’’ but not the ‘‘topological spin Hall effect’’ that is manifested by skyrmions through the real-space spin chirality. I would like to discuss this point further below...

... the term and phenomenon of the ‘‘topological spin Hall effect’’ seem to be not a widely accepted concept. The phenomenon caused by the emergent magnetic field due to the magnetic topology of skyrmions is the topological Hall effect, and I have never heard a phenomenon called the topological spin Hall effect. I think that most researchers have never heard that a phenomenon called the topological spin Hall effect which is illustrated in Fig.1a really occurs.”

Answer: As the reviewer pointed out, the THE can have contributions both from the real-space topology and that from the momentum-space topology that is caused by magnetic structures of the localized spins like skyrmions. The real-space contribution to the THE is usually considered in the adiabatic limit where the spin of the electron follows the direction of the local magnetization. In this limit, the THE was described as a spin separation process which arises from the opposite sign of the effective magnetic field (real-space Berry curvature) for opposite spins, e.g. by Bruno et. al in [PRL **93**, 096806 (2004)]. This leads to an asymmetrical spin-dependent scattering which was theoretically predicted to result in a spin current as described in Refs. [Yin et al. Phys. Rev. B **92**, 02441 (2015), Buhl et. al. Phys. Status Solidi RRL **11**,

1700007 (2017), Ndiaye et al. Phys. Rev. B **95**, 064426 (2017)]. In ferromagnets (FMs), spin separation of the uneven spin populations naturally leads to a charge Hall effect. Therefore, the THE and the so-called topological spin Hall effect (TSHE) coexist in FMs, making the term “TSHE” redundant and less common in this case. However, in antiferromagnets, the TSHE and THE do not necessarily coexist and the TSHE can arise without the THE. In this case, the term TSHE is more frequently used (see for example Buhl et al. Phys. Status Solidi RRL **11**, 1700007 (2017); Šmejkal et al. Nat. Physics, **14**, 766 (2018); C. A. Akosa et al. PRL **121**, 097204 (2018); Gobel et al. Eur. Phys. J. B **91**, 179 (2018)). In our work, a FM crystal is studied and the term THE is more appropriate. Therefore, in the revised manuscript the term TSHE has been completely removed.

Additionally, as the referee pointed out, the band topology or the Berry phase in the momentum space produced by the spin chirality of the localized spin structure can also cause the THE. In this case, the THE does not necessarily involve a spin current. To prevent ambiguity, we have revised the manuscript to explicitly distinguish between the real-space and k -space contributions to the THE wherever this distinction is relevant. Additionally, to avoid confusion, the schematic illustration appearing in the original Fig. 1(a) was removed.

We point out that in the non-adiabatic limit, the scattering mechanism is not spin-dependent and the THE does not involve spin currents as well [Denisov et al. Phys. Rev. Lett. **117**, 027202 (2016)]. This is discussed in the summary section.

Point 4

Current-driven dynamics of skyrmions in bulk materials is usually through spin-transfer torques. In this case, the skyrmions are driven by spin-polarized electric currents rather than spin currents.

Answer: Following the reviewer's remark, the role of STT was studied. To this end, we carried out additional micromagnetic simulations which incorporate the Zhang-Li STT term: $\propto \hat{m} \times (\hat{m} \times (\vec{u} \cdot \nabla) \hat{m})$ where \vec{u} is the Zhang-Li velocity term in the direction of the applied current (further details in the new ‘Methods’ section). The numerical calculations were carried out by merely replacing the SOT with an STT induced by the same current density, J_c . The study of STT pivoted around two key questions:

- 1) Whether STT can excite the AC skyrmion resonance dynamics.
- 2) Whether STT contributes to the broadening of the skyrmion resonance linewidth.

To examine the ability to excite the dynamics, an AC current was applied and the temporal response of the mean out-of-plane component of \vec{M} was calculated. Our results demonstrated that AC STT can also excite the dynamics. This persisted as long as the externally applied magnetic field (H_0) was kept below the saturation conditions before the texture was suppressed. As the externally applied magnetic field further increased and texture saturation took place, the simulations indicated that the efficiency of the STT-induced AC drive decreased as expected.

Interestingly, in contrast to DC SOT, we found that DC STT does not affect the skyrmion resonance linewidth in the stripes and lattice phases. The figure below shows the calculated skyrmion resonance linewidth broadening as a function of J_c for the STT case (panels (d)-(f)), alongside the original measured results (panels (a)-(c)). It is readily seen that STT does not induce a linewidth broadening. To validate that DC STT was indeed present, the calculated textures were examined as well (panel (g)). The influence of STT was observed only when texture switching took place above a critical J_c value which was larger than the J_c applied

experimentally. The textures corresponding to the highest J_c applied experimentally are highlighted in blue. In the disordered phase, STT-induced texture switching occurs at very low J_c , within the range of J_c applied experimentally (highlighted by the red frame).

Measured and calculated linewidth modulation in presence of DC STT. (a)-(c) Measured linewidth modulation $\Delta H_i - \Delta H_i^0$ at 5, 10, and 12 GHz. (d)-(f) Calculated $\Delta H_i - \Delta H_i^0$ for DC STT. (g) Magnetic textures in the presence of DC STT. Calculated textures as a function of J_c for different H_0 values. Blue highlighting indicates the maximum J_c applied experimentally. Red squares guide the eye to the regions where texture switching occurred.

Following this additional study, the claims regarding the excitation of dynamics by spin currents were revisited and the broader term 'spin-torque' which encompasses both SOT and STT is now used. Additionally, we have incorporated a new discussion of the role of STT and added the new simulations. Figure 3 was updated and now includes the linewidth calculations for the case of DC STT. Additionally, the new 'Supplementary Note 4' presents the calculated textures and texture switching for the DC STT case and the details of the STT calculations have been added to the 'Methods' section.

This additional study highlights the distinct ways in which STT and SOT influence the skyrmion resonance, leading to an important new conclusion. As a result, the abstract, introduction, and summary have been updated to reflect this point.

Point 5

In addition, it is certainly possible for Fe_3Sn_2 as a layered material to generate a vertical spin current due to the spin Hall effect caused by interfacial spin-orbit interactions. This vertical spin current could possibly drive the skyrmions by exerting the spin-orbit torque.

Answer: The mechanism whereby by interfacial spin-orbit interactions give rise to the spin Hall effect is now explained in the revised version of the manuscript and appears in the introduction.

Point 6

"...it is not clear what the authors mean when they claim that the spin current can be detected by measuring the magnetic resonance excitations using this experimental method. The origin of the magnetic resonance excitations is the oscillation of the skyrmions driven by the AC electric current, but the basis for the claim that this is a spin current or a spin current generated by the topological spin Hall effect due to the skyrmions, is unclear..."

"...Moreover, if such a phenomenon really occurs, it is not obvious at all how the spin currents generated via this phenomenon would be related to the magnetic resonance excitations observed experimentally in the present study the explanation required for such a claim is missing in the paper..."

Answer: As mentioned above, the claim that spin currents were generated via the TSHE which in turn drive the skyrmion resonance has been removed.

As for the ability to detect spin currents, in the original manuscript the spin currents were modeled by introducing a Slonczewski SOT. This was shown to result in a modulation of the skyrmion resonance linewidth as observed experimentally. Our additional study of STT showing that a DC STT does not modulate the resonance linewidth, further strengthens the assessment that the current induced skyrmion resonance linewidth modulation stems from an SOT-like process. In the revised manuscript we have separated between the experimental observations and the calculations from their interpretation. The discussion of the possible origins of the current-induced linewidth broadening was confined to the summarizing discussion. Additionally, the connection between the measurements DC transport measurements spin dynamics is presented now as a possibility rather than a definite claim.

Point 7

I believe that the significance of this paper lies in the two points mentioned above and that they are important results worth publication in Nature Communications. However, the

introduction part of this paper does not mention the significance of this paper. On the other hand, the authors' claim that the spin currents generated via the skyrmion-induced topological spin Hall effect can be detected via the skyrmion resonance measurements contains some errors and is not supported by firm argument. We think that active assertion of this claim is a major detriment to the value of this paper. In conclusion, we believe that if the entire paper including the introduction part is revised to correctly discuss the results and significance of this research, we may recommend this paper for publication in Nature Communications.

Answer: We thank the reviewer for their valuable feedback. We believe that the amendments made following their remarks have improved the quality of the manuscript. We hope the reviewer finds the revised manuscript satisfactory and that it adequately addresses their concerns.

Response to comments of Reviewer #2

Point 1

“The application of skyrmion resonance as a sensor for spin currents via the OSTFMR technique is innovative. Although spin-torque resonance methods are established, using them to probe skyrmion dynamics in a frustrated magnet like Fe_3Sn_2 is creative and may yield fresh insights into spin-torque effects in complex magnetic textures. By examining both the intrinsic spin Hall effect (iSHE) and the topological spin Hall effect (TSHE) in the same material, this study provides valuable insights into a system where both k-space and real-space chirality contribute to spin transport. The finding that iSHE is dominant while TSHE plays a minor role advances our understanding of spin currents in nontrivial magnetic textures and the interplay between iSHE and TSHE in frustrated systems.

The authors present consistent experimental results and micromagnetic simulations, offering a detailed analysis of skyrmion resonance modes and current-induced effects. Their quantitative extraction of parameters, such as effective spin Hall conductivity and Gilbert damping, enhances the rigor of the manuscript. Fe_3Sn_2 is a relatively rare example of a frustrated magnet with robust skyrmion stability at high temperatures and a unique independence from spin-orbit coupling (due to frustration rather than Dzyaloshinskii-Moriya interaction), distinguishing it from more conventional skyrmion systems. This work advances previous studies by investigating linewidth modulation under both AC and DC currents and exploring the anisotropic nature of spin-torque in frustrated systems.

While this study may not be a groundbreaking contribution to skyrmion research overall, as skyrmion dynamics and spin-torque effects are well-established, it adds meaningful insights specific to frustrated magnets and spin current detection. To broaden its impact, the authors could suggest potential applications or implementations of this technique, as well as possible extensions to other materials and practical devices.”

Answer: We thank the reviewer for the thorough examination of our manuscript. Per the reviewer's remark, we have revised our introduction and summary sections that now elaborate on the insights that may be gained from applying this technique to other materials and applications.

Point 2

“The research is technically sound, innovative, and well-suited for Nature Communications, though it may not be considered transformative. Its strengths lie in the novel application of OSTFMR to skyrmion resonance and the detailed analysis of spin Hall effects in Fe_3Sn_2 .”

Other suggestions:

Line 295: Change "100" to "[100]."

Figure 2 Caption: Change “Each response was normalized to the peak value.” to “Each response was normalized to its peak value.”

Abstract: Add error bars to the effective spin Hall conductivity for precision.

In the abstract and conclusion, it would be beneficial to emphasize the unique contributions of this research, such as insights into the anisotropic nature of spin-torque in frustrated magnets and advancements in skyrmion resonance as a sensor for spin currents.

Answer: The notation "100" was replaced by the more scientific "[100]" notation in the main text and in the 'Supplementary materials'.

The caption of Fig. 2 was corrected according to the reviewer's remark.

The effective spin Hall conductivity reported in the abstract now includes the estimated error.

Per the reviewer's comment, the abstract and concluding section emphasize also the insights gained into the anisotropic nature of the spin torque as well as the possibility of using the skyrmion resonance as a sensor of spin currents.

We thank the reviewer for their valuable feedback.